# Towards Cost-Efficient Federated Multi-Agent Reinforcement Learning with Learnable Aggregation

## Abstract

Multi-agent reinforcement learning (MARL) often adopts centralized training with a decentralized execution (CTDE) framework to facilitate cooperation among agents. When it comes to deploying MARL algorithms in real-world scenarios, CTDE requires gradient transmission and parameter synchronization for each training step, which can incur disastrous communication overhead. To enhance communication efficiency, federated MARL is proposed to average the gradients periodically during communication. However, such straightforward averaging leads to poor coordination and slow convergence arising from the non-*i.i.d.* problem which is evidenced by our theoretical analysis. To address the two challenges, we propose a federated MARL framework, termed cost-efficient federated multi-agent reinforcement learning with learnable aggregation (FMRL-LA). Specifically, we use asynchronous critics to optimize communication efficiency by filtering out redundant local updates based on the estimation of agent utilities. A centralized aggregator rectifies these estimations conditioned on global information to improve cooperation and reduce non-*i.i.d.* impact by maximizing the composite system objectives. For a comprehensive evaluation, we re-create a federated multi-agent autonomous driving environment based on MetaDrive. Our findings indicate that FMRL-LA can outperform other baselines by at least $5\%$ with respect to the system utility on average.

## 1 Introduction

Federated reinforcement learning (RL) (Chen et al., 2021; Khodadadian et al., 2022; Zhuo et al., 2019; Jin et al., 2022; Cha et al., 2020) has exhibited immense potential in integrating deep reinforcement learning models into a client-server paradigm. It has been proven effective in balancing communication efficiency and privacy preservation. With the burgeoning rise of the Internet of Things (Pinto Neto et al., 2023) that requires agent cooperation, and the prevalent use of multi-agent systems (MAS) (Lowe et al., 2017), it is desirable to develop federated multi-agent reinforcement learning (MARL) frameworks.

In MARL, centralized training with decentralized execution (CTDE) (Rashid et al., 2020; Kuba et al., 2022; Sunehag et al., 2017) is a conventional learning regime lying between independent learning (de Witt et al., 2020) and fully decentralized learning (Wen et al., 2022). This middle-ground strategy can not only mitigate the non-stationarity caused by agents' simultaneous decision-making, but also prevent state and action spaces from expanding exponentially with agent number. Nevertheless, the training phase of CTDE requires continual communication between agents and servers. Thus, simply incorporating CTDE into federated learning (FL) (McMahan et al., 2017) will lead to intractable communication overhead and bandwidth burdens. On the other hand, agent interactions with their local environments make their experiences non-independent and identically distributed (non-*i.i.d.*).

While recent efforts like FMARL (Xu et al., 2023) and Fed-MADRL (Song et al., 2022) have marked advances in federated MARL (Kumar et al., 2017; Chen et al., 2021; Li et al., 2022b), they typically assume an implicit *i.i.d.* in agent interactions and lack server-side coordination. Furthermore, these methods tend to optimize singular, task-oriented objectives, *e.g.,* the average speed in multi-vehicle

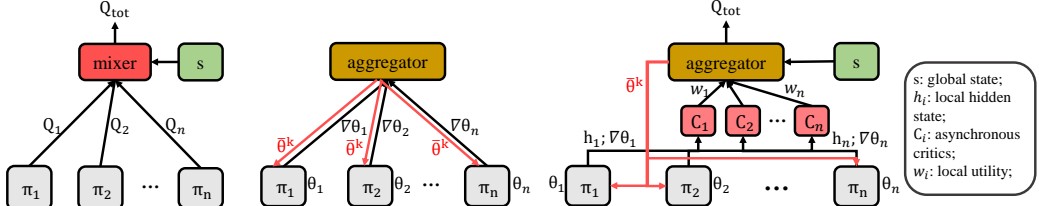

Figure 1: Framework Comparison. $\pi_i$: local policy whose parameters are $\theta_i$; $Q_i$: local value function; $Q_{tot}$: joint value function; $\nabla\theta_i$: local gradients; $\bar{\theta}^k$: global model's parameters at round k; $h_i$: local hidden state; $w_i$: local utility. The left figure is the conventional value decomposition framework where the parameters of the local policies are synchronized from time to time. The middle figure indicates one current federated MARL framework where local parameters are synchronized periodically and the global model is updated by direct averaging over local gradients. The right figure is our proposed method which follows the federated learning paradigm and the weights of local gradients for updating the global model are produced by the learnable aggregation module.

autonomous systems (Xu et al., 2023) and the system throughput in wireless communications (Song et al., 2022). Such settings may be impractical for complex real-world settings with composite objectives. For instance, in autonomous driving (Li et al., 2022a; Peng et al., 2021b), apart from communication efficiency, we also consider factors like success rate in reaching destinations, overall safety, and average vehicle speed.

In response to these challenges, we introduce Cost-Efficient Federated MARL with Learnable Aggregation (FMRL-LA). It decouples the CTDE by separating the training steps of the server and the client. On the server side, we propose two components for learnable aggregation: 1. Asynchronous critics evaluate the utility of learning agents, guiding selection for optimal system communication. 2. A centralized aggregator integrates global information with agent utilities to periodically update the global model, thus maximizing composite system targets. This design facilitates FMRL-LA to improve coordination under the federated paradigm. Delving deeper into the non-*i.i.d.* challenge posed by federated MARL, we theoretically delineate its adverse effects, providing a convergence upper bound. We further prove that the proposed learnable aggregation can mitigate the challenge. The comparison of different frameworks is exhibited in Fig. 1.

To conduct experiments with FMRL-LA, we resort to real-world multi-agent environment simulations based on MetaDrive (Li et al., 2022a), an intricate autonomous driving benchmark out of its flexibility across diverse scenarios. We extend it to support a client-server learning paradigm, incorporating communication efficiency. To further enhance the practicality, in addition to the existing navigation tasks, we design a multi-vehicle cooperative exploration task. Notably, we have integrated baselines from the representative methods of cooperative MARL (de Witt et al., 2020) and communication-inclusive MARL (Foerster et al., 2016), as well as the state-of-the-art method (Peng et al., 2021b) using MetaDrive. Our experimental evaluations in navigation and exploration tasks underscore that FMRL-LA can optimize system performance and efficiency simultaneously, delivering a balanced performance across the metrics corresponding to composite objectives.

## 2 PRELIMINARY

**Cooperative MARL** Cooperative MARL can be formulated as Decentralized Partially Observable Markov Decision Processes (Dec-POMDPs) (Rashid et al., 2020; Kuba et al., 2022), described by a tuple $G = \langle n, S, O, A, P, r, Z, \gamma \rangle$, where $n$ is the number of agents, and $S$, $O$ denote the state and observation spaces. $A$, the joint action space, is the product of all agents' action spaces, *i.e.,* $A = \prod_{i=1}^{n} A_i$, where $i$ is the agent index. We use lowercase $s$, $o$, $a$ to represent an element in the corresponding space. The environments' dynamics are characterized by the transition function $P(s'|s, a) : S \times A \times S \to [0, 1]$. The system has a shared team reward function $r(s, a) : S \times A \to \mathbb{R}$. In the aspect of each agent, due to the partially observable setting, at time step $t$, its observation $o_t$ is drawn by applying the function $Z$ to the current state $s_t$. Thus, $o_i^t = Z_i(s^t) : S \to O$. $\gamma$ is the discount factor. The solution of a Dec-POMDP is a joint policy $\bar{\pi} = (\pi_1, \pi_2, ..., \pi_n)$, where $\pi_i$ stands for the policy of agent $i$ and we use $\theta_i$, $\bar{\theta}$ to represent the parameters of agent $i$ and the joint policy, respectively. Each agent policy is trained with the agent's experience comprised of a

collection of agent observation-action history denoted as $\xi_i = \{(o^t, a^t, o^{t+1})\}_{t=0}^T$, where T denotes the time horizon. In addition, we use $\xi = \{(s^t, a^t, s^{t+1}, r^t)\}_{t=0}^T$ to represent one global team episode. The goal of MARL is to learn a joint policy that can maximize the expected cumulative reward, *i.e.,* $\pi^* = \arg\max_{\bar{\pi}} \mathbb{E}_{\tau \sim \bar{\pi}}[R_T(\tau)]$, where $R_T(\tau) = \sum_{t=0}^T \gamma^t r^{(t)}$.

**Federated MARL**   We use $\tau$ to represent the number of local updates. K is the termination condition of the training process, which is usually set as maximum communication rounds (Chen et al., 2021). $\psi$ denotes the system communication efficiency. We use the parameter $\theta$ to represent policy $\pi$. $F(\cdot)$ represents the global objective function, while $F_i(\cdot)$ stands for the local objective function for each agent $i$. Their relationship between the global objective and the locals in (Song et al., 2022; Xu et al., 2023; Chen et al., 2021) are the same: $F(x) = \frac{1}{n}\sum_{i=1}^n F_i(x)$. In round $k$, all agent policies are synchronized as $\bar{\theta}^k$, which is drawn from the server. Then, each agent interacts with the environment concurrently to accumulate local experience for updating the local policy indicated by $\{\theta_i^{k,\tau_i}\}_{i=1}^n$. Next, the parameters $\{\theta_i^{k,\tau_i}\}_{i=1}^n$ or stochastic gradients $\{g(\theta_i^{k,j}; \xi_i^{k,j})\}_{j=1}^{\tau_i}$ for $i \in 1, 2, \cdots, n$ will be uploaded to the server. To sum up, the update rules on the server and client side are:

$$\bar{\theta}^{k+1} = \bar{\theta}^k - \eta \frac{1}{n} \sum_{i=1}^n \sum_{j=1}^{\tau_i^k} g(\theta_i^{k,j}), \quad \theta_i^{k+1,j} = \begin{cases} \bar{\theta}^{k+1}, & j \bmod \tau_i = 0, \\ \theta_i^{k,j} - \eta g(\theta_i^{k,j}), & \text{otherwise.} \end{cases} \quad (1)$$

To indicate the convergence of the algorithm, we use the expected averaged gradient norm to guarantee convergence to a stationary point (Wang et al., 2020; Bottou et al., 2018; Wang & Joshi, 2021):

$$\mathbb{E}[\frac{1}{K} \sum_{k=0}^{K-1} ||\nabla F(\bar{\theta}^k)||^2] \leq \epsilon, \quad (2)$$

where $|| \cdot ||$ is the $\ell_2$-norm and $\epsilon$ is used to describe the sub-optimality. When the above condition holds, we say the algorithm achieves an $\epsilon$-suboptimal solution.

## 3   FEDERATED MARL WITH LEARNABLE AGGREGATION

**Server Side**   When federated MARL (Xu et al., 2023; Song et al., 2022) adopts Eq. 1 as the update rule for the server, it implicitly assumes that the agents are homogeneous. However, in real-world environments, the agents are diverse in various aspects such as computing capability, network connection, and local observation distributions, which results in heterogeneous agents with non-*i.i.d.* experience distribution.

To deal with these issues, we introduce **Asynchronous Critics** to dynamically evaluate the agent utilities in each communication round. Each critic corresponds to one learning agent. Its goal is to maximize the return of the current agent. The inputs are hidden information $h_i^k$, accumulated rewards in recent communication round $r^k := \sum_{j=\tau_i^{k-1}}^{\tau_i^k} r_i^j$ and in agent history $\sum_{j=0}^{\tau_i^k} r_i^j$. The output is a prediction of the agent's local utility:

$$w_i^k = C_i\left(h_i^k, r^k, \sum_k r^k\right), \quad (3)$$

where $C_i$ is the asynchronous critic network of agent $i$. The output $w_i^k$ can be zero, which means the corresponding agent does not need to upload its training parameters to the server in the current communication round to implement client selection.

Next, the agent utilities are passed through a **Centralized Aggregator** to facilitate coordination. It works similarly to the mixing network in value decomposition methods such as (Rashid et al., 2020; Sunehag et al., 2017), which takes the local utility function as input and facilitates agent coordination by maximizing the system utility condition on the global state. The RL loss is back-propagated to the critics for improving the local utility estimation:

$$Q_{tot} = Mix\left(w_1^k, w_2^k, \cdots, w_n^k\right), \quad (4)$$

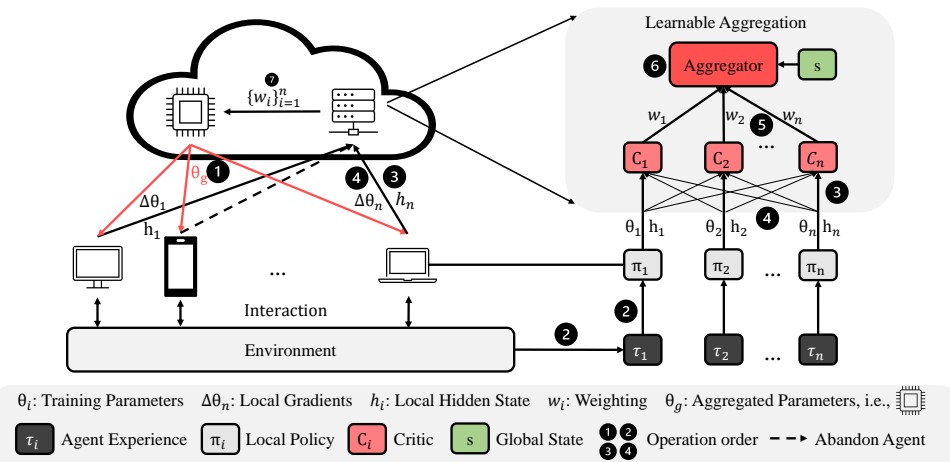

Figure 2: The workflow of our proposed framework.

where $Q_{tot}$ denotes the system utility, which is reflected by the composite objectives. The server aggregates the gradients based on the local utilities to update the global policy. Thus, the update rule of the server is:

$$\bar{\theta}^{k+1} = \bar{\theta}^k - \eta \sum_{i=1}^{n} w_i^k \sum_{j=1}^{\tau_i^k} g(\theta_i^{k,j}), \qquad (5)$$

while the update rule for the clients remains the same as conventional federated MARL.

**Client Side** Considering the generalization of our method, we choose an independent reinforcement learning algorithm and take the hidden states as additional outputs. During the communication, the upload process of the clients can be divided into two stages. In the first stage, the agents upload their rewards and hidden information to the asynchronous critics for local utility estimation and agent selection, optimizing the communication efficiency. In the second stage, the selected agents upload their gradients to the server for aggregation.

**Framework Design** Compared to the update rule for FMARL (Xu et al., 2023; Song et al., 2022) represented by Eq. 1, we adopt a weighted aggregation for global policy update implemented by the learnable aggregation module. The workflow of FMRL-LA is illustrated by Fig. 2. Specifically, 1. the server broadcasts the global model $\bar{\theta}^k$ to each agent; 2. The agents learn local behavior policies $\{\pi_i\}_{i=1}^{n}$ by interacting with the environment and maintaining hidden states; 3 and 4. During client-server communication, agents conduct the two-stage upload described in the above subsection; 5. The centralized aggregator maximizes the composite system objectives condition on the global states to facilitate coordination; 6. The global model is updated based on the local utilities.

## 4 CONVERGENCE ANALYSIS

In FL theory, a substantial body of research is devoted to exploring convergence properties under diverse settings. These settings predominantly fall into two categories, *i.i.d.*(McMahan et al., 2017; Xu et al., 2023; Song et al., 2022; Chen et al., 2021), and non-*i.i.d.* (Li et al., 2020; Wang et al., 2020; Karimireddy et al., 2020). While *i.i.d.* settings facilitate robust theoretical results, non-*i.i.d.* settings are more realistic about data distribution. Despite the theoretical progresses in FL schemes in supervised learning, the influence of non-*i.i.d.* in federated MARL remains uncharted. To address this issue, we conduct our theoretical analysis in the following paragraphs.

We begin with showing the convergence under the ideal *i.i.d.* setting. To do that, we first list out the following assumptions:

**Assumption 1** *(Lipschitz continuity) The local loss functions at the client side are Lipschitz continuous, which means* $||\nabla F_i(\theta_1) - \nabla F_i(\theta_2)|| \leq L||\theta_1 - \theta_2||, \forall i \in \{1, 2, ..., n\}$.

**Assumption 2** *(Unbiased gradients and bounded variance under i.i.d.) The stochastic gradients at the client side are unbiased estimators of the global gradient, i.e., $\mathbb{E}_\xi[g_i(\theta)] = \nabla F(\theta)$ and $\mathbb{E}_\xi[||g_i(\theta) - \nabla F(\theta)||^2] \le \mu||\nabla F(\theta)||^2 + \sigma^2, \forall i \in \{1,2,...,n\}$, $\mu$ and $\sigma^2$ are non-negative.*

**Assumption 3** *(Unbiased local gradients and bounded variance under non-i.i.d.) The stochastic gradient at each client is an unbiased estimator of the local gradient, i.e., $\mathbb{E}_\xi[g_i(\theta)] = \nabla F_i(\theta)$ and $\mathbb{E}_\xi[||g_i(\theta|\xi) - \nabla F_i(\theta)||^2] \le \mu||\nabla F_i(\theta)||^2 + \sigma^2, \forall i \in \{1,2,...,n\}$, $\mu$ and $\sigma^2$ are non-negative.*

**Assumption 4** *(Bounded Dissimilarity) For any sets of weights $\{w_i^k \ge 0\}_{i=1}^n, \sum_{i=1}^n w_i^k \le M^k, M^k$ is finite, $\forall k \in [0,K]$, there exist constants $\beta^2 \ge 1, \kappa^2 \ge 0$ such that $\sum_{i=1}^n w_i^k||\nabla F_i(\theta)||^2 \le \beta^2 \{||\sum_{i=1}^n w_i^k \nabla F_i(\theta)||^2, ||\sum_{i=1}^n \frac{1}{n} \nabla F_i(\theta)||^2\}_{min} + \kappa^2, \forall k \in [0,K]$. If local loss functions are identical to each other, then we have $\beta^2 = 1, \kappa^2 = 0$.*

Assumption 1 is Lipschitz continuity, a common assumption in the convergence analysis in FL theory. Assumption 2 states that the local stochastic gradient is an unbiased estimation of the local gradient and the variance of the deviation is bounded to support our exploration under a *i.i.d.* setting. Assumption 3, on the other hand, is the gradient bias and variance assumption under non-*i.i.d.* setting. Assumption 4 is inspired by FedNova Wang et al. (2020), which bounds the dissimilarities on the weighted norm of local gradients.

We provide the convergence bound under the *i.i.d.* and non-*i.i.d.* settings, respectively. In Theorem 3, we show that the learnable aggregation mechanism can potentially reduce such impact. The prove of these theorems as well as more theoretical details are provided in Appendix A.3.

**Theorem 1** *Suppose $\{\theta_i^{k,j}\}$ and $\{\bar{\theta}^k\}$ are parameters' sequences generated by equation 1. The federated MARL is conducted under Assumptions 1 and 2. If the total communication rounds K is large enough, which can be divided by $\tau$, and the learning rate $\eta$ satisfies:*

$$\{L\eta < 1, 2L^2\eta^2\tau(2\mu + 1 + \tau) < 1\}, \tag{6}$$

*then the expected gradient norm after K iterations is bounded by:*

$$\mathbb{E}[\frac{1}{K}\sum_{k=1}^K ||\nabla F(\bar{\theta}^k)||^2] \le \frac{2[F(\bar{\theta}^1) - F(\bar{\theta}^K)]}{\eta K} + \frac{\eta L\sigma^2}{n} + \eta^2 L^2 \sigma^2(\tau + 1), \tag{7}$$

*where $\bar{\theta}^1$ stands for one lower bound of the objective function.*

**Theorem 2** *Suppose $\{\bar{\theta}^k\}$ are parameters' sequences generated by the weighted gradients equation 2, while the $\{\theta_i^{k,j}\}$ remains the same. The federated MARL is conducted under Assumptions 1, 3 and 4. If the total communication rounds K is large enough, and the learning rate $\eta$ satisfies equation 6, then the expected gradient norm after K iterations is bounded by:*

$$\mathbb{E}[\frac{1}{K}\sum_{k=1}^K ||\nabla F(\bar{\theta}^k)||^2] \le \frac{4\left(E\left[F\left(\bar{\theta}^1\right) - E\left[F\left(\bar{\theta}^K\right)\right]\right]\right)}{K\eta}$$

$$+ 4\left(C + D + E + F + \mu\eta C \sum_{k=0}^K \frac{1}{K}\sum_{i=1}^n w_i^2 \tau_i^k\right), \tag{8}$$

*where $\bar{A} = \frac{1}{K}\sum_{i=1}^K A$, $B = 2L^2\eta^2\tau(2\mu + 1 + \tau)$, $C = \frac{\eta^2\sigma^2 L^2}{\mu L\eta\tau\beta^2 + 2B\beta^2}$, $D = \frac{(1-2\mu L\eta\tau\beta^2)\kappa^2}{(2\mu L\eta\tau\beta^2 + 4B\beta^2)(1+4\beta^2)}$, $E = \frac{\mu L\eta\tau\kappa^2}{2\mu L\eta\tau\beta^2 + 4B\beta^2}$, and $F = \frac{L\eta\sigma^2}{2K}\sum_{k=1}^K (M^k)^2$;*

**Theorem 3** *Suppose the same condition as Theorem 2, we can reduce the convergence upper bound by tuning the aggregation weights. If we define $w_i \to \frac{1}{n\tau_i^k}$, then the expected gradient norm after K iterations is bounded by:*

$$\mathbb{E}[\frac{1}{K}\sum_{k=1}^K ||\nabla F(\bar{\theta}^k)||^2] \le \frac{4\left(E\left[F\left(\bar{\theta}^1\right) - E\left[F\left(\bar{\theta}^K\right)\right]\right]\right)}{K\eta}$$

$$+ 4\mathcal{O}\left(\bar{A} + C + D + E + F + \mu\eta C \sum_{k=0}^K \frac{1}{K}\sum_{i=1}^n w_i^2 \tau_i^k\right). \tag{9}$$

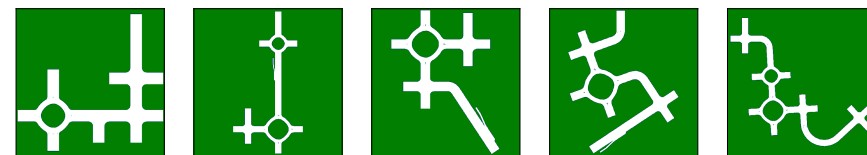

Figure 3: The six extended scenarios used in our evaluation.

## 4.1 DISCUSSION

The result of Theorem 1 is an ideal upper bound where the distribution of each client is *i.i.d.* More generally, in Theorem 2, we provide another upper bound to illustrate the impact introduced by the non-*i.i.d.* issue. In Theorem 3, we prove that learnable weighting can reduce such impact.

**Special Cases** When $w_i^k \equiv \frac{1}{n}$, the convergence upper bound degenerates to the same as FMARL (Xu et al., 2023), which derives the same upper bound as in its Theorem 2. When $w_i^k \equiv \frac{1}{n}$ and $\tau_i^k \equiv 1$, the convergence upper bound further degenerates to the same as PASGD, which coincides with the conclusions drawn from (Wang & Joshi, 2021).

**Discussion with Federated Learning in supervised learning** We compare our method with FedNova (Wang et al., 2020) – a general federated method targeting supervised learning. It induces several federated learning methods in a general form and targets the problem of an unbalanced number of local updates by regularizing the weights for one-period local gradients with the number of local updates. However, in MARL, the different number of policy iterations may not be a more significant reason than the diversity of local environments to the non-*i.i.d.* issue. In other words, this issue cannot be rectified by simply regularizing the weights of local gradients by the number of local policy iterations, which necessitates the importance of our learnable aggregation mechanism.

**System Utility** To provide a comprehensive evaluation for fair comparisons, we propose a general utility function that reconciles both theoretical and practical considerations. Specifically, from the perspective of numerical performance, both task-oriented performance and system efficiency are crucial metrics. We denote them as $Q_s$ and $\psi$, respectively. For instance, in a multi-agent navigation environment, $Q_s$ can be a composite objective comprised of navigation success rate, average speed, and safety while $\psi$ refers to the system communication and computation cost. As for the convergence property, we consider an ideal setting where the agents are *i.i.d.* It serves as the optimal convergence upper bound $\epsilon_m$. By comparing the convergence bound with it, we can tell the tightness of the federated MARL method. Thus, we derive our system utility function as $Q_{tot} = \frac{Q_s - \lambda \psi}{e^{\|\epsilon - \epsilon_m\|^2}}$, where $\lambda$ is a positive constant used to balance the importance of system cost and performance.

## 5 EXPERIMENTS

**Baseline Methods** We present a comparative analysis of our proposed method alongside strong baselines from related fields, namely conventional MARL (**IPPO** (de Witt et al., 2020)) and communication-based MARL (**RIAL and DIAL** (Foerster et al., 2016)), as well as state-of-the-art methods **CoPO** (Peng et al., 2021b), **FMARL** (Xu et al., 2023) in a multi-agent autonomous driving simulation environment, MetaDrive (Li et al., 2022a). We provide a detailed introduction and adaptation of these methods in the Appendix A.5.

**Implementation by Extending MetaDrive** The MetaDrive benchmark is a flexible, lightweight autonomous driving simulation benchmark that contains several tasks. In this paper, we focus on its multi-agent tasks. The agents adopt the conventional MARL suite, including parameter sharing and disregarding communication overhead. We add six challenging scenarios whose maps are depicted by Fig 3. Originally, MetaDrive used parameter sharing for all the methods, so we first expanded this benchmark into a client-server learning setting by adopting a non-parameter sharing scheme and simulating a virtual server. This server only periodically collects local gradients and hidden states from the clients, aggregates the gradients for the update of the global model, and then sends it back to the agents. When an existing vehicle terminates and a new vehicle spawns, it accepts the latest global model from the server to prevent the "cold start" problem. To enrich the testing bed, in addition to the existing navigation task, we extend a cooperative exploration task where vehicles cooperatively explore the specified destinations.

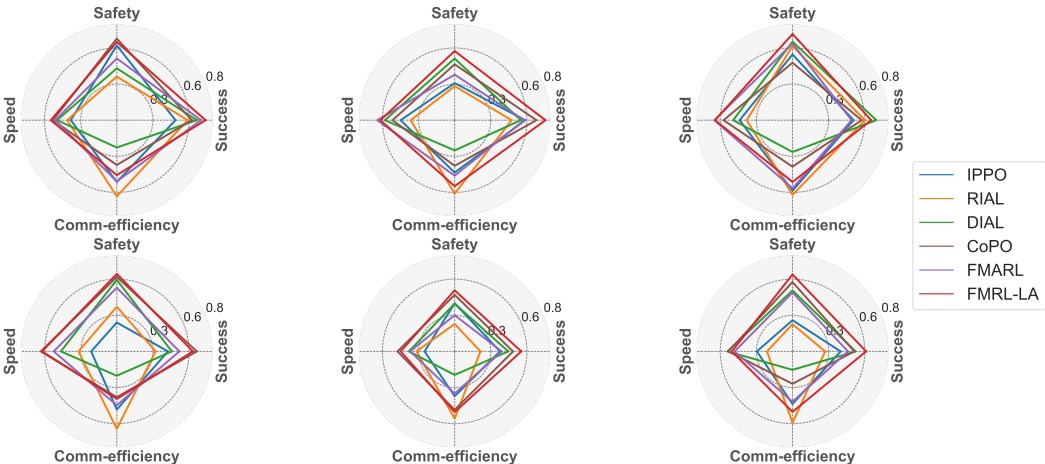

Figure 4: The system performance and efficiency comparison with baselines in six scenarios of the cooperative navigation tasks.

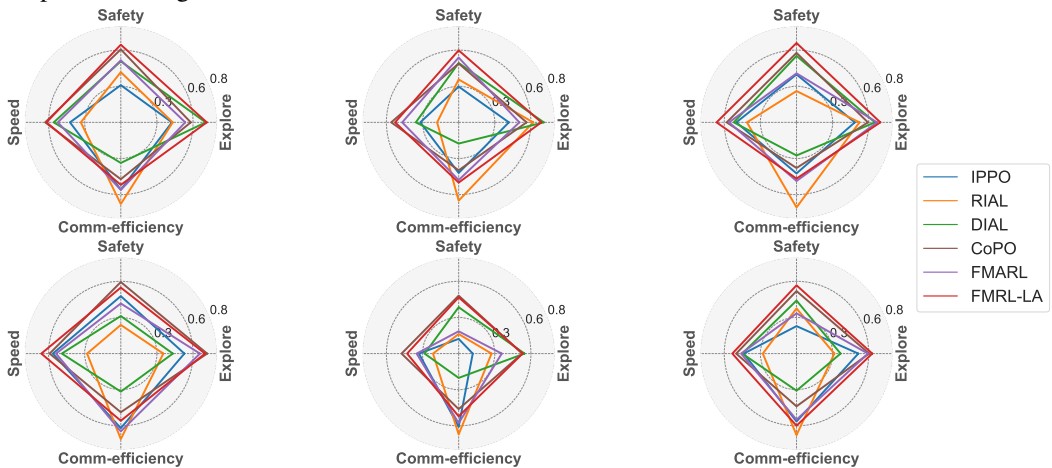

Figure 5: The system performance and efficiency comparison with baselines on six scenarios of the cooperative exploration tasks.

**Evaluation Metrics**   MetaDrive provides three evaluation metrics: success rate, efficiency, and safety, which respectively reflect navigation capability, navigation efficiency, and safety driving. In our cooperative exploration task, we adapt the navigation success rate to the exploration success rate. For realistic concerns, we also record the communication overhead which is reflected by the number of parameters exchanged between the agents and the server.

In summary, in the **cooperative navigation** task, our evaluation metrics including the navigation success rate (*Success*), safety (*Safety*), overall agents' speed (*Speed*), and communication efficiency (*Comm-efficiency*). As for the **cooperative exploration**, our evaluation metrics including the exploration success rate (*Explore*), safety (*Safety*), overall agents' speed (*Speed*), communication efficiency (*Comm-efficiency*).

**Main Results Analysis**   The experiment results on cooperative navigation and exploration across six scenarios can be found in Fig. 4 and Fig. 5, respectively. More detailed results related to the performance on two tasks can be found in Tab. 2, 3, 4, and 5 in the Appendix A.6.

**Our Performance.** In both tasks, though the preference for different metrics relies on the specific environments, we use the average of the metrics as the system utility for fair comparison. We find that FMRL-LA achieves or is comparable to the best success, speed, safety, and system utility. Since more than half of the baselines adapt PPO(Schulman et al., 2017) as the algorithm of the clients, while RIAL (Foerster et al., 2016) uses a simpler algorithm with fewer parameters, the communication efficiency of our method cannot outperform it. But compared to the baselines with the same client-side

algorithm, our method exhibits the capacity to dynamically harmonize the system performance and communication efficiency.

In detail, we focus on the performance of IPPO, FMARL, and our FMRL-LA in both tasks. The three methods have nearly the same client-side algorithms but differ from each other on the server side. IPPO only conducts direct training parameter averaging, while the FMARL adds a weight decay mechanism during the averaging. And FMRL-LA dynamically learns the aggregation weights. From IPPO to FMARL, then to FMRL-LA, the performance of success, safety, speed, and system utility follow an ascending manner. We believe that it is because the performance of IPPO is bound by the averaging capability of all learning agents while FMARL can enlarge the bound to some extent by weight decay. Nonetheless, the potential performance of FMRL-LA is bound by the best agent, which improves the generalization of our method since we can deploy a suitable behavior model for the clients in advance if we can make use of prior knowledge about the environments.

**Task Comparison.** Comparing the overall performance of all methods on cooperative navigation and cooperative exploration on six scenarios, we can find navigation is more difficult than exploration, especially in relatively complex scenarios, *e.g.*, scenario 4, 5, and 6. We notice that in the navigation task, each agent has its own destination. Therefore, we believe the different performance on the two tasks may be because the relationship between the local utilities and the system utility is easier to capture in exploration than in navigation.

**Scenario Difficulty.** In both tasks, if we compare the performance pair by pair such as scenario 1 and scenario 5, we observe that generally, the more building blocks involved in a scenario, the more challenging it is. Then, looking into the performance of scenario 1 and 2, both of them consists of four building blocks while scenario 2 contains one more roundabout than scenario 1. If we compare the safety of CoPO, FMARL, and our FMRL-LA, three robust methods in the two tasks on these six scenarios, we can find that roundabout tends to result in more crushes. Further, if we compare the performance of scenarios 1 and 4 on two tasks, we observe that though the two scenarios both contain one roundabout and the same number of building blocks, the performance of our method and other baselines is generally better in scenario 1. Considering the difference in these two scenarios, we hypothesize that it is due to the influence of wide turn. For intuitive methods like IPPO and RIAL, it is difficult for them to avoid crushing or driving out of the roads during the wide turns. On the other hand, the safety of CoPO in scenarios 3 and 4 is relatively high, it may benefit from its explicit modeling of the surrounding agents.

**Ablation Study**    To investigate the effectiveness of our design and components in FMRL-LA, we conduct an ablation study about the usage of asynchronous critics and a centralized aggregator as well as an alternative design for the federal mixer. From Tab. 1 we can observe that if we directly use critics without the centralized aggregator, the performance is unstable, resulting in large standard errors. In scenario 4, the performance w.r.t. system utility is worse than federated IPPO. We believe that without the coordination of the centralized aggregator, the server cannot filter out less valuable agents, so their parameters can depreciate the update of the global model in the current round. Meanwhile, the asynchronous critics are useful in our method since the variant that only uses an aggregator performs worse than the full model. We believe that accepting information from all involved agents can stagnate the learning of an aggregator due to redundant information. When we change the centralized aggregator to a VDN-based (Sunehag et al., 2017) one, it yields an inferior performance compared to our QMIX-based (Rashid et al., 2020) design, which suggests the non-linear modeling of the relationship between the agents and the server is more suitable for complex realistic environments than a simple sum as in VDN.

**Client Selection Analysis**    To investigate the effectiveness of the learnable aggregation on the perspective of client selection during client-server communication, we conduct experiments on cooperative navigation tasks in scenarios 1, 2, and 3 with different numbers of learning agents. The results are shown in Fig. 6. The average number of selected agents is calculated during the evaluation intervals, which is consistent with the calculation of all the evaluation metrics. Thus, it reflects the number of agents selected for the overall training phase. The less the agents are selected, the higher the communication efficiency we achieve. When there are only 3 agents in the scenarios, due to the partial observability and the complexity of the tasks, the agents require a longer time to learn stable policies. Besides, each agent's gradients are important to the system. Thus, the learnable aggregation module does not select agents' gradients frequently and may pay more attention to weighting the gradients toward a better collective policy learning. However, as the number of agents increases in

Table 1: Ablation study on the effectiveness of our critical components on navigation task. Average system utility is provided.

| Experiment | Scenario1 | Scenario2 | Scenario3 | Scenario4 | Scenario5 | Scenario6 |
|---|---|---|---|---|---|---|
| IPPO | 49.96±3.06 | 45.13±3.68 | 51.49±2.73 | 33.12±5.29 | 34.41±6.17 | 34.33±5.97 |
| FMRL-LA w/o aggregator | 54.75±6.00 | 49.28±5.23 | 52.06±4.71 | 40.43±7.02 | 32.85±8.92 | 40.22±7.74 |
| FMRL-LA w/o critics | 52.69±3.55 | 48.46±4.65 | 55.49±3.18 | 45.35±5.26 | 38.49±6.19 | 48.07±5.57 |
| FMRL-LA w/ vdn-aggregator | 57.98±2.97 | 55.84±4.39 | 57.26±3.69 | 52.50±5.80 | 46.98±6.81 | 50.86±5.09 |
| FMRL-LA | **59.84**±2.82 | **62.30**±4.29 | **63.16**±3.33 | **57.27**±4.70 | **50.28**±6.67 | **56.42**±5.63 |

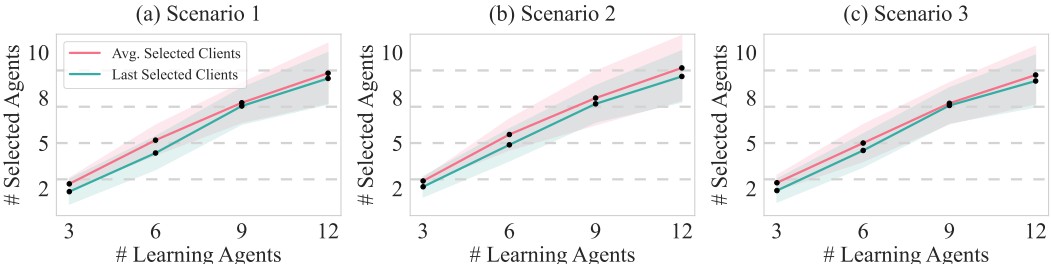

Figure 6: Number of selected clients under different numbers of learning agents for the three scenarios

the same tasks, the effect of selection becomes more significant. More experiments related to the results of selection effects can be found in Tab. 6 in Appendix A.6.

# 6 RELATED WORK

**Cooperative MARL**   Cooperative MARL has widespread applications in real-world scenarios (Yu et al., 2022a; Li et al., 2022a). Current methods are mainly developed in game scenarios (Rashid et al., 2020; Lowe et al., 2017) where the methods can focus on technical design rather than practical details. These environments support parameter sharing (PS) (Christianos et al., 2021) and CTDE regime (Lowe et al., 2017; Hu et al., 2021) to enable multiple agents to be trained on one device and facilitate cooperation, respectively. However, when it is the stage to consider practical MARL in realistic environments(Peng et al., 2021b; Abegaz et al., 2023), either PS or CTDE cannot be simply applied due to privacy concerns and communication overhead.

**Federated MARL**   Federated MARL (Song et al., 2022; Li et al., 2022b) appears to be a feasible way towards realistic MARL. Most of these methods enable agents to learn individual behavior policies and set a virtual server to maintain a global policy. The agents' policies are aggregated and updated periodically through communication with the server (Chen et al., 2021; Xu et al., 2022). In this way, the communication overhead is reduced, and the majority of them aggregate the local gradients by direct averaging (Xu et al., 2023) or weighted by the relative mini-batch size (Song et al., 2022). Although this oversimplified update may work well under *i.i.d.* setting, the MASs are naturally non-*i.i.d.* due to the interaction among agents. The notorious non-*i.i.d.* issue can stagnate convergence (Li et al., 2020; Wang et al., 2020; Li et al., 2019). Besides, without centralized training, it is hard for MARL to learn coordination (Kuba et al., 2022; Fu et al., 2022).

# 7 CONCLUSION

We aim to adapt MARL for real-world applications by introducing a hybrid distributed, client-server learning framework that takes into account communication and computation overhead. Our framework offers theoretical guarantees even under the influence of non-*i.i.d.* distribution of agents in local environments. To empirically validate the efficacy of our proposed Cost-Efficient Federated Multi-Agent Reinforcement Learning with Learnable Aggregation (FMRL-LA) method, we modify an existing multi-agent autonomous driving simulation environment to conform to a client-server scheme. Experimental results emphasize the superior performance against baseline methods.

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
