# OpenReview forum: "Towards Cost-Efficient Federated Multi-Agent Reinforcement Learning with Learnable Aggregation"
_ICLR.cc/2024/Conference — ICLR 2024 Conference Withdrawn Submission_

### Official Review · Reviewer_PGsv · 2023-10-31

**Soundness:** 3 good
**Presentation:** 3 good
**Contribution:** 2 fair
**Rating:** 5
**Confidence:** 3

**Summary:**

This work proposes FMRL-LA for federated MARL training to reduce the communication overhead between the server and the training clients (agents). It utilizes asynchronous critics and a centralized aggregator to optimize communication efficiency and improve cooperation among agents, showing improvement in system utility compared to other baselines in a recreated federated multi-agent autonomous driving environment.

**Strengths:**

- The paper is relatively easy to follow
- The proposed method seems to be able to get good performance compared the to provided baseline.

**Weaknesses:**

- The proposed technique is only tested on a single environment, is it also applicable to other environments? Does the problem of periodic gradient averaging problem have a significant effect on the performance?
- This work seems to focus on the non-i.i.d issue of the MARL, but in the evaluation section, most of the compared baselines are not designed for a non-i.i.d setting. How about techniques from federated reinforcement learning?
- The proposed method is not compared to the case where there is frequent communication (i.e. centralized training), what is the performance and communication trade-off ?

**Questions:**

- The proposed method seems to outperform the existing method in terms of existing methods. How does it compare to centralized training with full communication? How much is the performance degradation?
- Having communication between agents seems to be a different dimension. In the evaluation section, the proposed method is compared to baselines with (e.g. DIAL, RIAL) and without communication between agents, but the proposed technique is only tested on cases without communication between agents. Would you get better performance if you included inter-agent communication?
- How does the degree of non-i.i.d affects the performance of MARL, at what point does it requires your proposed technique?

---

### Official Review · Reviewer_XrBt · 2023-11-01

**Soundness:** 2 fair
**Presentation:** 2 fair
**Contribution:** 2 fair
**Rating:** 3
**Confidence:** 4

**Summary:**

The paper addresses the cost of communication and computation overhead in federated multi-agent reinforcement learning.
The authors propose a hybrid distributed, client-server learning framework, FMRL-LA, that introduces a linear aggregation layer on top of the distributed learning in the federated paradigm.
The framework aims to improve communication efficiency and coordination among agents while reducing the impact of non-i.i.d. distribution of agents in local environments.
The authors provide a theoretical analysis of the proposed method in both i.i.d. and non-i.i.d. settings, which includes convergence results of the aggregated parameter on the server side.
Experiments are conducted using customized tasks by extending the MetaDrive environments to validate the results.

**Strengths:**

The paper introduces a novel framework aimed at improving communication efficiency in the complex domain of federated multi-agent reinforcement learning. This objective is achieved by transforming the cooperative multi-agent reinforcement learning (MARL) problem into a distributed/federated reinforcement learning setup with a server-client architecture.

The inclusion of convergence results for such a federated reinforcement learning problem in non-iid (non-independent and identically distributed) settings is noteworthy, although I did not thoroughly assess the rigor of the proofs provided.

The paper's evaluations and ablation studies appear comprehensive, providing a deeper understanding of the proposed FMRL-LA framework.

**Weaknesses:**

This raises some confusion for me about whether the primary focus of interest lies in multi-agent reinforcement learning, as mentioned at the beginning of Section 2, or if it is more aligned with distributed or parallel reinforcement learning:

- For instance, if all agents operate within a single environment, I'm curious about how, during step 2 in the FMRL-LA framework, each agent learns its local behavior policy. Does the environment transition to the next timestep after receiving a single action from any agent, or does it wait for a vector of actions from all agents before transitioning? Is the system clocked synchronously?

- On the other hand, if agents are operating in distributed environments, I'm interested in understanding whether it is still considered a cooperative multi-agent problem. How does the setup in this paper differ from distributed or federated reinforcement learning approaches such as [1,2]?



The paper bears the title "Cost Efficiency," yet it lacks clarity regarding the extent of this efficiency. I highly recommend that the authors incorporate a thorough analysis and discussions concerning the costs, particularly in comparison to related works.



In terms of technique, it appears that the primary distinction between the proposed FMRL-LA and the existing FMARL framework lies in the introduction of the linear aggregation layer. I would like to suggest that the authors delve into a discussion regarding the significance of FMRL-LA as a technical contribution.


Another potential weakness is that the evaluation is limited to the customized tasks extending the MetaDrive environment. While this is a relevant and challenging problem domain, it would be beneficial to evaluate the proposed framework in other domains to demonstrate its generalizability.


The notation used in Figure 2 is not consistent with that in the main text of the paper.

I would be pleased to revise my assessments once any misunderstandings have been resolved.



[1] Chen, Y., Zhang, X., Zhang, K., Wang, M., & Zhu, X. (2023, April). Byzantine-robust online and offline distributed reinforcement learning. In International Conference on Artificial Intelligence and Statistics (pp. 3230-3269). PMLR.

[2] Fan, X., Ma, Y., Dai, Z., Jing, W., Tan, C., & Low, B. K. H. (2021). Fault-tolerant federated reinforcement learning with theoretical guarantee. Advances in Neural Information Processing Systems, 34, 1007-1021.

**Questions:**

Please see the questions in weaknesses.

---

### Official Review · Reviewer_Vyxa · 2023-11-06

**Soundness:** 3 good
**Presentation:** 3 good
**Contribution:** 2 fair
**Rating:** 5
**Confidence:** 4

**Summary:**

Aiming at adapting  MARL for real-world applications, this paper introduces a novel framework, Cost-Efficient Federated Multi-Agent Reinforcement Learning with Learnable Aggregation (FMRL-LA), which addresses the challenges of communication efficiency and non-independent and identically distributed (non-i.i.d.) experiences in Multi-Agent Reinforcement Learning (MARL). Specifically, asynchronous critics were used to optimize communication efficiency by filtering out redundant local updates based on the estimation of agent utilities, and a centralized aggregator rectifies these estimations conditioned on global information to improve cooperation and reduce non-i.i.d. impact by maximizing the composite system objectives. It was claimed that the proposed FMRL-LA can optimize system performance and efficiency simultaneously, delivering a balanced performance across the metrics corresponding to composite objectives. Some theoretical analysis are provided, as well as some simulation/empirical evidence.

**Strengths:**

Strengths:

The problem studied in this paper is interesting and could be beneficial to important real world MARL applications.

The paper provides valuable theoretical analysis and claims that FMRL-LA can mitigate the non-i.i.d. challenge existed in previous existing federated MARL literature.

The paper provides empirical evidence of the effectiveness of FMRL-LA by modifying an existing multi-agent autonomous driving simulation environment, showing it outperforms baselines by at least 5% with respect to the system utility on average.

**Weaknesses:**

Weakness:

The proposed FMRL-LA framework is tested only in a specific use case (autonomous driving), limiting the generalization of the results. I'd like to see more comprehensive benchmark comparison with existing state-of-the-art methods to more convincingly claim the proposed new framework is significantly superior to existing methods, for broad representative practical MARL applications.

The paper does not provide a comprehensive comparison with other existing solutions to the non-i.i.d. problem in federated MARL. It would be helpful if the authors could provide comprehensive benchmark performance comparison between existing algorithms (e.g., FMARL) and the proposed FMRL-LA on broad non-iid task scenarios, understand how significant/insignificant of the non-iid impact on more representative MARL applications, and evaluate how much advantage FMRL-LA has and whether it consistently demonstrates significant gains compared with existing methods.

Compared with existing literature (e.g., FMARL), the contribution/novelty in this work seems incremental (though still valuable), and it is not sufficiently convincing that the contribution is significant enough for a regular paper at such a top tier conference like ICLR.

**Questions:**

see weakness section comments.